# Validation of the DigCompEdu Check-in Questionnaire through Structural Equations: A Study at a University in Peru

Lorena Martín Párraga * , Carmen Llorente Cejudo and Julio Barroso Osuna

Department of Didactics and Educational Organization, University of Seville, 41013 Sevilla, Spain
* Correspondence: lorena@grupotecnologiaeducativa.es

**Abstract:** The technologization of society presents a great challenge for education in the twenty-first century; there is a need to face that challenge to be able to promote quality digital literacy. The use of the teacher's digital competence in terms of the safe and critical use of technologies is one of the key competencies that can guarantee educational success. The present study analyzes the reliability and validity of the DigCompEdu questionnaire for future teachers within the framework of digital competence improvement. This tool is centered on teacher orientation, with respect to their level of competence, through a self-evaluation of their strengths and needs for improvement in digital learning, according to its different dimensions: digital literacy, communication and collaboration with the organization; search and treatment of data, digital socialization; technological creativity and innovation. For its development, the exploratory and confirmatory factorial analysis (EFA and CFA) technique was utilized via structural equation modeling (SEM). A total of 1659 professors, who are employed at a university in Peru, completed the questionnaire. The analyses that were performed corroborated the reliability and validity of the instrument, as well as the different possibilities guaranteed by the validation method via structural equations. We underline the need to offer training for professors in the area of digital competence and endorse methodologies based on competencies that guarantee the use of valid and reliable tools.

**Keywords:** digital technologies; digital teaching competence; competency framework; structural equations model; active methodologies

## 1. Introduction

Today's society is immersed in a multitude of constant changes, due to the progressive use of information and communication technology, which has forged a new technological era. This technological transformation has resulted in important advances at the social, economic, and cultural levels of society, not forgetting the subject of our concern, the area of education. The easy, immediate access provided to each of the sectors that compose this cycle points to the necessary ingredients that will provide an added value for the improvement of society, in terms of knowledge. These predictions indicate that the fourth industrial revolution demands new digital skills to secure future employment [1].

This technologization of society is the reason why new modifications are made in terms of the organization of information, knowledge, and ways of communicating, as well as the modeling of human cognition.

These accelerated changes have an effect on many areas, among which we find the teaching profession, given the difficulties experienced by educators when they try to update their knowledge to adapt it to the vertiginous rhythm of technology. This versatility, provided by the incorporation of ICT, has resulted in educational institutions bringing forward the updating of their methodological plans, given the inclusion of these technologies, to offer them a place in their educational practices [2,3].

Universities, to a greater extent, must face the challenge of investigating new manners to promote the teaching–learning (T-L) process, considering the alterations produced in the present society [4–6].

The additional use of ICT in teaching is problematic in the face of the traditional type of teaching that is predominant today. Thus, there is a need for the updating and training of educators according to the changes experienced by this new reality, with the acquisition of key competencies being essential; these competencies will equip educators with the knowledge and guidelines needed for the effective use of ICT in teaching.

Therefore, it is understood that there is a need for the digital training of educators to guarantee the acquisition of key competencies, defined as "a combination of knowledge, skills, and attitudes associated with the context" [7] (p. 7). It is, therefore, essential to acquire the associated competencies in a way that is able to respond to current demands. Thus, focusing on the present state of affairs, and according to what was outlined by the European Union's Council, we can state that one of the most important competencies in this technological era is that of digital competency, involving the reliable and essential use of the new wave of technologies for work, enjoyment, and communication [7].

Based on the above, in 2012, the European Commission planned to "redirect education" as a means to attain quality education in the current environment of transformation, granting usefulness to and integrating ICT into the learning processes in an efficient manner. This implies the development of international training plans that are able to effectively incorporate digital competencies among educators, implying a common standard of education in this competence [8].

This technologization has been able to transform literacy practices, acquiring a fundamental role in the appropriate development of present educational contexts. Therefore, the need arises to review the concept of literacy and to make advancements in terms of new ways of identification, facilitating greater access to the development of competencies that are socially expected, offering a digitized culture that is able to reveal digital literacy, e-learning, e-inclusion, e-health, and commitment to digital solutions [9] (p. 4). The importance of facilitating diverse digital platforms and the technological and didactic educational resources to educational systems are also evidenced as guarantors of the correct use of ICT during the T-L process.

Given the demands of the digital era and the need to acquire a broad sample of competencies and strategies, a list of necessary skills has been created by official institutions and organizations, with digital competency (DC) being found in all of them [10]. For many authors, the term "digital competency" alludes to the creative, critical, and safe use of technologies as tools for achieving the correct performance of work, academic, or leisure objectives, and even for active integration and participation in society [11–13].

The importance of developing correct digital literacy as a tool for knowing how to use, manage, evaluate, and identify the ICT has been reported previously [14]; it is fundamental in the search and treatment of data [15] and the development of critical thinking that allows for the resolution of problems and the making of correct decisions [16,17]. On the other hand, the acquisition of essential skills for the development and implementation of digital strategies that are oriented toward collaboration and the communication of information [18] leads to the establishment of ethical notions through good practice [19], with a subsequent effect on the deployment of more innovative and creative educational practices [20]. Even so, from a teaching perspective, it has been suggested that immersion in this technological current does not ensure equal opportunities for its access and use, causing possible social inequalities and, as indicated by the authors of [21], may generate visible inequalities in the different levels of competence. There is a need to turn teachers into content generators, creators of ideas and opinions, relieving them of the passive mindset generated by the lack of teacher training in this field [21]. This approach will foster the relationship between competencies and the adoption of teaching methodologies among teachers since, as indicated in the studies carried out by the authors of [22], who analyzed the teachers' perspective, the higher the level of digital competence, the greater the teachers' willingness to incorporate e-learning modalities. In turn, as proficiency improves, more favorable changes will be generated in the didactic models used for the integration of ICT in teaching and learning processes [23].

As the training of university professors in DC is viewed as an urgent necessity [24], many national and international institutions have begun to work on frameworks and models that will facilitate digital competence [25]. Due to this, the National Institute of Education Technologies and Teacher Training developed a reference framework for the diagnosis and measurement of digital competencies of professors, in order to deal with this technological barrier. Thus, in 2017, with the intention of achieving the acquisition of a reference framework by the European Policies, the Joint Research Center of the European Union presented the European Framework of Digital Competence for Educators (DigCompEdu), the result of numerous studies conducted at the local, national, European, and international levels [26,27].

DigCompEdu constitutes a competency model of six areas (see Table 1) incorporating the different competencies that education professionals must develop to promote productive, inclusive, and integrative learning strategies through the use of digital tools [28], as described by the authors of [29].

**Table 1.** DigCompEdu areas of competence.

| Areas | Competencies |
|---|---|
| Area 1 | Professional commitment, centered on the importance of the educator's work environment. |
| Area 2 | Digital resources, in agreement with the creation and distribution of digital resources. |
| Area 3 | Digital pedagogy, one of the essential competencies within the framework. This is focused on the creation, organization, and implementation of the ICT in the T-L process. |
| Area 4 | Evaluation and feedback: associated with the use of digital resources and strategies for evaluation. |
| Area 5 | Empowering students, which instills the importance of the correct use of appropriate digital tools for empowering students in their learning. |
| Area 6 | Student competencies: related to the educator's capacity when facilitating DC among the students. |

Within the framework, different levels of competency are also established (see Table 2), comprising a total of 6 progressive levels of mastery. This scheme was created for better detection of the educator's competencies, making possible the gradual personal development and autonomy. It starts with an initial level A1, to a more complete C1.

**Table 2.** Different DigCompEdu competence levels.

| | Levels of Mastery |
|---|---|
| A1 | The person possesses a basic level of competence, which requires support for its future development. |
| A2 | The subject has acquired a basic level of competence, which, with adequate support, will lead to an improvement in DC. A certain independence has also been achieved during its practice. |
| B1 | The person possesses a medium level of competence, being able to solve simple problems and to gradually make progress toward the development of DC. |
| B2 | The person has an intermediate level of competence but is now able to provide answers to his or her needs and to solve correctly defined problems, with definite progress observed in the development of his or her competence. |
| C1 | The subject has a more advanced level of competence, which means that he or she is able to guide other individuals toward an increase in digital competence. |
| C2 | The person has reached an advanced level of competence, being able to meet his or her needs, just as they can meet the needs of others. The subject has developed a level of competence that is able to provide answers to complex situations. |

The "DigCompEdu" model was created to develop a self-reflection tool for educators, known as the "DigCompEdu Check-in", which is based on the European Framework of Digital Competence for Educators. The main objective of the questionnaire is for educators to improve their comprehension of said framework, by providing them with a self-evaluation of their strengths and weaknesses, which is needed for educators to become highly competent in their professional practice.

Once the questionnaire is completed, the tool itself is responsible for creating a detailed and personalized report on the level of competence, according to areas of mastery. This instrument is oriented towards different education stages, focusing on the stage of interest to us, university educator training.

The instrument is composed of 22 items that include the content from the 6 areas of competence established in the common framework of digital educator competence: profes-

sional commitment (1), digital resources (2), digital pedagogy (3), evaluation and feedback (4), empowering students (5), and facilitating the digital competence of students (6) [29].

The content of the DigCompEdu Check-in has been included in the self-assessment tool, EuSurvey; thanks to its global classification system, sorted according to area, this allows us to identify the level of digital competence acquired by an educator. For this study, a graduated classification system will be used to discover the global digital competence of educators.

## 2. Materials and Methods

The purpose of the present study is linked to an analysis of the DigCompEdu questionnaire. To measure the reliability and validity of the instrument, exploratory factorial analysis (EFA) and confirmatory factorial analysis (CFA) were performed through the use of structural equation models (SEM).

The SEM techniques allow us to analyze how the existing covariance is distributed in each type of data, and to evaluate if the relationships between variables, as expressed through the model, adjust to the values [30]. In summary, it is a procedure that consists of defining a conceptual model, which represents the relationships between a set of latent factors and their observed variables, from which a covariance matrix will be obtained that will be compared with the matrix from the SEM to measure the validity of this model [31].

### 2.1. Sample

The total study population was composed of 1659 university educators employed at the Continental University of Peru. Of these, 568 (34.3%) were women and 1090 (65.7%) were men. Most of the respondents to the "DigCompEdu Check-in" questionnaire, which was developed by the European Framework of Digital Competence for Educators to measure the level of educator competence, were aged between 30 and 39 years old (34%) and between 40 and 49 years old (34.9%).

### 2.2. Data Collection Instrument

For the collection of data and the posterior analysis of the data, the "DigCompEdu Check-in" questionnaire [29] was used, an analysis instrument generated by the European Framework of Digital Competence for Educators (DigCompEdu), validated by Ghomi and Redecker (2018). This framework was selected because it is fundamental in the assessment of DC in university educators through its validation via the expert judgment technique [32].

This instrument is composed of twenty-two items, which are distributed into the six areas of competence analyzed in DigCompEdu. These are related to the different areas of competence: (A) professional commitment (4 items), (B) digital resources (3 items), (C) teaching and learning (4 items), (D) assessment (3 items), (E) empowering students (3 items), and (F) facilitating the DC of students (5 items).

By means of this questionnaire, teachers were asked initially to self-assess their level of competence, thereby classifying themselves in one of the following categories: novice explorer, leader, or pioneer. The same process was repeated once the questionnaire was completed, to verify its level of significance.

In addition, a series of demographic questions were included in the questionnaire, covering sex, age, years of service, and time spent using technologies, among others.

### 2.3. Collection and Analysis of Data Procedures

The administration of this questionnaire was performed digitally through EuSurvey at the end of 2021; the questionnaire was distributed to university personnel from a Peruvian university, the method guaranteeing the anonymity of the data.

To measure the validity of the questionnaire, reliability and validity studies were performed on the instrument and on the information obtained, for a high level of scientific rigor.

To calculate the reliability and the discriminant and convergent validity, the following coefficients were considered: Cronbach's alpha, McDonald's omega, composed reliability (CR), average variance extracted (AVE), and maximum shared variance (MSV), chosen according to the studies conducted by the authors of [32]. So that the obtained results could be contrasted, an inferential statistics analysis was conducted between the items and dimensions, to ensure systematic and efficient evaluation. For this purpose, a bivariate correlation analysis was performed using Spearman's *p*-correlation coefficient.

To test the validity of the construct, an exploratory factorial analysis (EFA) was utilized, using the principal components method with Varimax rotation and Kaiser normalization. Once the factors were obtained, confirmatory factor analysis was performed (CFA), to verify if the theoretical means of the model had a good internal consistency, through the use of structural equations [30]. Thus, it was verified that the data obtained did not have a normal distribution through their descriptive study, in which asymmetry and kurtosis were taken into account. To verify this, a Kolmogorov–Smirnov goodness-of-fit test was performed, obtaining a significance equal to 0.000 for the totality of the items.

The program utilized for the analysis of data was JASP 0.16.2. Eric-Jan Wagenmakers (sala G 0.29), University of Amsterdam, Amsterdam, the Netherlands.

## 3. Analysis and Results

The data obtained were subjected to an analysis of reliability through the calculation of Cronbach's alpha, with values close to 1 indicating the reliability of its scales [33], along with McDonald's omega, applied globally and for all the dimensions that constituted the questionnaire.

The data collected obtained a Cronbach's alpha of 0.937 globally (Table 3). It was determined that the index was very high (> 0.9), which signifies a high degree of reliability. Table 4 shows the reliability indices according to the dimensions from the questionnaire: professional commitment (0.813), digital resources (0.755), digital pedagogy (0.978), assessment and feedback (0.863), empowering students (1.08), and facilitating the digital competence of the students (0.914).

**Table 3.** Global Cronbach´s alpha.

| Reliability Statistics | |
| --- | --- |
| Cronbach´s Alpha | Number of elements |
| 0.950 | 22 |

**Table 4.** Reliability dimensions.

| Reliability Statistics | | | |
| --- | --- | --- | --- |
| Cronbach´s Alpha | | Number of elements | |
| 0.933 | | 6 | |
| | Mean | Standard Deviation | *N* |
| A | 2.1578 | 0.81384 | 808 |
| B | 2.3333 | 0.75536 | 808 |
| C | 2.0619 | 0.97881 | 808 |
| D | 1.7814 | 0.86309 | 808 |
| E | 1.8639 | 1.08937 | 808 |
| F | 1.8594 | 0.91461 | 808 |

Each of the values, as determined by the authors of [34], obtained levels of reliability that were higher than > 0.75 for the instrument as a whole, as well as the different dimensions that it comprises; therefore, we considered that each dimension possessed a high level of reliability.

Next, the simple correlations of each item, with its theoretical dimension, were calculated. The results are shown in Table 5.

**Table 5.** Correlation of the items with the associated dimensions.

| Matrix of the Component [a] | |
|---|---|
| | Component 1 |
| A1 | 0.634 |
| A2 | 0.580 |
| A3 | 0.627 |
| A4 | 0.649 |
| B1 | 0.625 |
| B2 | 0.587 |
| B3 | 0.504 |
| C1 | 0.757 |
| C2 | 0.755 |
| C3 | 0.714 |
| C4 | 0.783 |
| D1 | 0.796 |
| D2 | 0.747 |
| D3 | 0.762 |
| E1 | 0.735 |
| E2 | 0.753 |
| E3 | 0.736 |
| F1 | 0.667 |
| F2 | 0.768 |
| F3 | 0.700 |
| F4 | 0.758 |
| F5 | 0.791 |
| Extraction method: analysis of principal components. | |

[a] 1 Extracted components.

Each of the values that were higher than 0.5 was considered, which allows us to accept the item as a component of that dimension [35].

To calculate the validity of the construct, an EFA analysis was performed (Table 6). Prior to this, its applicability was confirmed through the sampling adequacy test KMO and Bartlett's sphericity test. The results showed KMO values with a statistically significant coefficient of 0.977, very close to 1, which indicates a high degree of association between the items, along with a sphericity test of 31038.347 in the chi-square, with a *p*-value < 0.000 (see Table 6). In summary, the factorial analysis of the data could be applied.

**Table 6.** KMO and Bartlett´s test.

| Kaiser-Meyer-Olkin Measurement of Sampling Adequacy | | **0.977** |
|---|---|---|
| Bartlett's sphericity test | Chi-square value | 31,038.347 |
| | gl | 231 |
| | Sig | 0.000 |

From this first analysis, we can extract factor 1, which explains 49.797% of the total variance. The method for the extraction used the principal components with Varimax rotation, from which we obtained the matrix of rotated components shown in Table 7.

**Table 7.** Method of extraction: analysis of the principal components.

| Total Explained Variance | | | | | | |
|---|---|---|---|---|---|---|
| Component | Initial self-values | | | Sum of the loads to the square of the extraction | | |
| | Total | % Variance | % Accumulated | Total | % of Variance | % Accumulated |
| 1 | 10.955 | 49.797 | 49.797 | 10.955 | 49.797 | 49.797 |

The model found by the EFA was contrasted with the CFA. For this, a global adjustment was made through the use of different statistical tests: chi-square (Cmin), goodness-of-fit index (GFI), parsimonious goodness-of-fit index (PGFI), normed fit index (NFI), and parsimonious normed fit index (PNFI).

Table 8 shows the values obtained and the reference values for the adjustment of the model, according to Lévy Mangin et al. (2006).

**Table 8.** Goodness-of-fit indices of the model.

| Index | Result | Fit | Good fit |
|---|---|---|---|
| CMIN | 338.347 | CMIN $\leq$ 500 | Yes |
| CFI | 0.994 | GFI > 0.7 | Yes |
| PGFI | 0.993 | PGFI > 0.7 | Yes |
| NFI | 0.993 | NFI > 0.7 | Yes |
| PNFI | 0.898 | PNFI > 0.7 | Yes |

Considering the indices obtained, it has been confirmed that the model is adequate and fits perfectly with the empirical data. Likewise, the results obtained also confirm the validity of the construct, thus allowing us to corroborate the statement that the model is pertinent for achieving the objectives defined in the study.

## 4. Discussion and Conclusions

The results of the present study are associated with the validity of the DigCompEdu questionnaire. The reliability and the validity of the instrument provide us with the possibility of creating rigorous, precise, and valid knowledge to offer regarding quality education in the present context of transformation. This is evidenced by the high indices of reliability, the questionnaire's theoretical validity, and its confirmatory structure.

According to the fit indices, the model is valid and fitted to the empirical data [36]. Thus, the validity and pertinence of the model can be confirmed. This validation described the existing reality, explaining the educators' perception of the importance of the subject analyzed, as well as the importance of its applicability for their future professional development.

The justification for validating the measurement instruments is supported by authors such as Cole and Maxwell [37], who attest to the relevance of being able to partially but firmly validate the measurement values to affirm both the precision of the data with which we worked and the metric properties of the instruments utilized during the process of research. In this way, we would be assured of maintaining a rigorous scheme. Along with this surety, the need to sustain a theoretical plan and its methodological design was also considered, as precision of the data was required to obtain a robust instrument.

On the other hand, when alluding to the nature of the subject addressed, and following other studies centered on this field, the importance of the referential frameworks that were planned is underlined, as well as the recommendation of how to replicate their study methodology with these models [38], as performed throughout the conducted study. Authors such as [39] point out that the competency models are considered "education priorities from each country, with a convergent view that easily formulates quality and inclusive education, and in which public policies facilitate the democratization of knowledge" as cited in [40,41] and according to different studies discussed throughout this article [14,18,20], it can be verified that all the information is not merely theoretical, but that an increasing number of different paths exist with respect to the digital materials created.

Previous works by [42] stress that almost all the studies conducted have the same weakness: that of assessing educators only in terms of their work in the classroom, ignoring their professional commitment to the community, aside from maintaining a certain pejorative view about the taxonomy of the teaching profession, and not considering the more holistic aspects of their work. At the same time, other studies show that teachers' competencies are not as wide-ranging as expected, showing the infrequent use of ICT in their educational practices [43], as well as the self-perception received by them based

on their low mastery of ICT, which generates insecurity when it comes to adapting their resources, due to their low levels of creation of digital content [44]. Other research [45] offers relevant results regarding the low levels of creation and adaptation of these contents among teachers, which is a worrying situation. This creates a new concept in terms of the need to consider the context in which the profession will be developed, as well as the capacities necessary when acquiring a comprehensive DC of teacher quality. With respect to the training of teachers, it has been verified that in all the education systems, there is a search for the constant and continuous development of the DC model for its standardized integration into educational institutions. This is why the establishment of levels of competence will lead to the establishment of more personalized training itineraries [46].

Along with these initiatives, the support, certification, and recognition of the public administrations will be in only the early stages and will require systematic, previous, and continuous analysis of the education plans [47].

On the other hand, it is important to emphasize the limitations of this study and to detail the fact that these are, to a greater extent, delimited by the characteristics of the sample. In future studies, it would be advisable to expand or replicate it in other national and international universities, in order to take into account the possible digital divide existing in the different geographical areas. However, the questionnaire has been created to serve as a template for better progress in the development of measurement instruments and can always be adapted to serve the needs of each educational center, creating a model of digital development that is capable of guiding educational policies at all levels, regardless of their technological wealth. On the other hand, it would be convenient, depending on the characteristics of the educational institutions under study, to identify the improvement in some ways that would offer more congruent results.

In conclusion, the development of digital competence is essential in this new virtual education, a change that has come to stay. For this development, it is necessary that educators become aware of their responsibility and assume the role of carriers of new pedagogic models, being aware of the importance of being up-to-date by receiving good training that will allow them to examine and innovate educational transformations in the future.

As detailed above, it is no longer enough to possess basic training; however, this training must be put into practice by including pedagogic actions that are able to lead to better performance regarding the use of ICT in the field of education, along with the importance of knowing how to assess the process that guarantees its correct progression.

Therefore, the application of this type of questionnaire, aimed at reinforcing and establishing a better mastery of digital competencies, will be crucial in the future, not only to improve teacher knowledge but also as an infallible means to self-evaluate the level of mastery that a teacher has achieved, as established by the Joint Research Center of the European Union in 2017 [26,27]. Even if one has a certain level of competence in handling digital technologies, this will not be enough to provide quality teaching in the use of technological media since, as has been detailed throughout this research, to achieve a correct inclusion, it will be necessary that the teacher is able to perform a reflective professional practice wherein the ability to produce content, share learning experiences, and be able to transform knowledge is enhanced, thereby contributing simultaneously to a construction of the teacher's own training that is aimed at the creation of their professional identity.

Society as a whole moves forward, taking on the work of continuous renovation, due to the importance of finding an equilibrium that guarantees future advancement and progress.

**Author Contributions:** L.M.P., C.L.C. and J.B.O. presented and designed the experiments; L.M.P. performed the experiments, analyzed the data, and wrote the original article. C.L.C. and J.B.O. contributed to the review and editing. All authors have read and agreed to the published version of the manuscript.

**Funding:** Design: production and evaluation of t-MOOC for the acquisition of digital compe-tences of university teachers. Reference: US-1260616. Junta de Andalucía (Consejería de Economía y Conocimiento).

**Institutional Review Board Statement:** Ethical review and approval for this study was waived because the subjects participating in the study (1659) responded to the signed consent form before answering the questionnaire.

**Informed Consent Statement:** Informed consent was obtained from all subjects involved in the study.

**Data Availability Statement:** Due to confidentiality and privacy agreements, it is not possible to make these data publicly available.

**Conflicts of Interest:** The authors declare no conflict of interest.

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
