# Peer review of "Validation of the DigCompEdu Check-in Questionnaire through Structural Equations: A Study at a University in Peru"

_education, doi:10.3390/educsci12080574_

Round 1

Reviewer 1 Report

The article provides a relevant study in the field of research. Although it should improve some aspects:

- The final discussion needs to be made more robust. Tying it to the literature at the beginning would benefit the reader and solidify the findings.

Author Response

Thank you for your comments. I am glad to know that you find the study relevant. I have been looking at your annotations, and it is true that this one should be a little more specific, in certain sections and contrast its importance with what is described throughout the text. I am reattaching the conclusions with the extra section included:

Reviewer 2 Report

Thank you for the well-readable, well-structured, and interesting article. I have only a few remarks and suggestions:

- The topic is widely discussed internationally (also Engl.) (--> Introduction and theoretical Background)  

- Is it only a self-assessment/-reflection tool or also a competency test? This is not sufficiently explained. How does the test work?

- 22 items for six competency areas is not much. Perhaps explain again briefly. 

- The limitations seem very tight. For example, the clear gender imbalance is not addressed. In general, the limitations should be dealt with more intensively. It remains questionable how meaningful the specific sample is for broad use in education. 

Author Response

(The authors gave the same response as above.)

Reviewer 3 Report

This is a very interesting paper in which authors study the digitalization of the society. This topic is one of the most studied in the last decade and the results shown in this study can validate the questionnaire.

There are just two things to be considered:

- The questions used must be presented in a clear way and in its present forms we cannot make an assessment.

- I urge authors to add more bibliography related to the use of technology by teachers and its perception, as there are in the last 2 years lots of results that must considered in the introduction and in the concluding section.

Author Response

Thank you for your comments. I am glad to know that you find the study relevant. I have been looking at your annotations, and it is true that this one should be a bit more specific, contrasting its importance with what is described throughout the text. I reattach the conclusions with the extra section included.
We attach the full text in the introduction and conclusions to facilitate their review, although we have only underlined the modifications in red.
